# *Escherichia coli* Isolated from Vegans, Vegetarians and Omnivores: Antibiotic Resistance, Virulence Factors, Pathogenicity Islands and Phylogenetic Classification

Ariane Tiemy Tizura [1],*[iD], João Gabriel Material Soncini [1][iD], Vanessa Lumi Koga [2], Renata Katsuko Takayama Kobayashi [2][iD], Zuleica Naomi Tano [3][iD] and Eliana Carolina Vespero [1][iD]

1    Department of Pathology, Clinical and Toxicological Analysis, Center for Health Sciences, State University of Londrina, Londrina, Paraná 86038-350, Brazil
2    Department of Microbiology, Center for Biological Sciences, State University of Londrina, Londrina, Paraná 86057-970, Brazil
3    Department of Medical Clinic, Center for Health Sciences, State University of Londrina, Londrina, Paraná 86038-350, Brazil
*    Correspondence: tiemytizura@gmail.com

**Abstract:** Pathogenic strains of *Escherichia coli* have acquired virulence factors, which confer an increased ability to cause a broad spectrum of enteric diseases and extraintestinal infections. The aim of this study was to analyze the antimicrobial resistance profile of and the presence of virulence-associated genes (VAGs) in *E. coli* fecal isolates from omnivores, vegetarians and vegans. A control group of 60 isolates from omnivores, as well as a study group with 41 isolates from vegetarians and 17 from vegans, were analyzed. Isolates from both groups showed a high rate of resistance to ampicillin, amoxicillin-clavulanic acid and nalidixic acid, and some of them were positive for the ESBL test (12% of isolates from vegetarians/vegans and 5% of isolates from omnivores). The most predominant VAGs detected in isolates from omnivores were *fim*H (70%), *iut*A (32%), *fyu*A (32%) and *tra*T (32%), while among isolates from vegetarians or vegans, the most predominant were *traT* (62%), *kpsMT* k1 (28%) and *iut*A (22%). Most isolates from omnivores (55%) were positive for PAI I536, while most of those from vegetarians/vegans (59%) were positive for PAI IV536. Phylogenetic group A, composed of commensal non-pathogenic isolates that survive in the intestinal tract, was the most prevalent in both control and study groups. Some VAGs were found in only one of the groups, such as the pathogenicity island PAI III536, found in 12% of the isolates from omnivores, while the *kpsMT* III gene (15%) was detected only among isolates from vegetarians/vegans. Interestingly, this gene codes for a polysaccharide capsule found mainly in *E. coli* isolates causing intestinal infections, including EPEC, ETEC and EHEC. Finally, our results show that there were no advantages in vegetarian or vegan diets compared to the omnivorous diet, as in both groups we detected isolates harboring VAGs and displaying resistance to antibiotics, especially those most commonly used to treat urinary tract infections.

**Keywords:** *Escherichia coli*; virulence-associated genes; pathogenicity islands; phylogenetic classification; vegetarians; antimicrobial resistance

## 1. Introduction

*Escherichia coli* is a member of the normal gut microbiota of healthy humans, but can also be an important human pathogen, capable of causing a variety of infections [1]. The pathogenic *E. coli* strains are subdivided into two main groups: diarrheagenic *E. coli* (DEC) and extraintestinal pathogenic *E. coli* (ExPEC). DEC cause diarrheal syndromes that vary in clinical presentation and pathogenesis, being distinguished in variants such as Enterotoxigenic *E. coli* (ETEC), Enteropathogenic *E. coli* (EPEC), Enterohemorrhagic *E. coli* (EHEC), Enteroinvasive *E. coli* (EIEC) and Enteroaggregative *E. coli* (EAEC) [2,3].

Molecular studies demonstrate the distinction of ExPEC from other *E. coli*, being pathogenically versatile and of great concern for human health. The ExPEC group is involved in severe infections at multiple anatomical sites, such as urinary tract infections (UTIs), pneumonia, meningitis and sepsis [4–6]. ExPEC is generally transmitted via the fecal–oral route and its entry into the human host from animal reservoirs can be straightforward, such as direct contact or consumption of foods made from animals, or more complicatedly, involving not only the original animal reservoir, but also other animals and environmental factors, being a pathogen of concern in relation to One Health issues [7]. The latter is defined as an integrative effort of multiple disciplines working locally, nationally and globally to achieve optimal outcomes through integrative management of human, animal and environmental health [8].

Bacterial resistance, as well as virulence factors, have become urgent topics among health professionals and institutions around the world. This is because the genes encoding resistance and virulence can be present in plasmids, easily transferred from one microorganism to another. This is even more so in the case of ExPEC, which can be present in the intestinal microbiota and dynamically interact with the external environment in a bidirectional manner, that is, bacteria can circulate among ecosystems: from animals to humans, through manure, feces, water and soil, returning to humans and animals by food and feed [8,9].

There is scant information in the literature on a possible role of animal-source food as a cause of developed resistance in bacteria, but there is growing concern about this contribution on the observed resistance to antibiotics in humans [10], since antimicrobials are prescribed in the pig, cattle and chicken industries, and also in the farmed seafood industry [11].

Considering that one of the main community infections caused by ExPEC is UTIs, epidemiological studies have suggested that there is an association between high pork and chicken meat intake and resistance to the main antibiotics used for UTI treatment, such as ciprofloxacin, ampicillin and third-generation cephalosporin [12]. In addition, several studies have shown similarities between *E. coli* extended-spectrum β-lactamase genes in broilers, retail chicken and human clinical isolates [13]. In addition, the human gut microbiota change according to different eating habits, and health depends on microbial metabolism within this community, although the influence of different lifestyles and diets on the microbiota composition is still not fully understood [14].

To investigate the possible influence of different diets on in the antimicrobial resistance profile of *E. coli*, people from a city in southern Brazil, Londrina, in the north of the State of Paraná, who eat different diets (omnivores, vegetarians and vegans) were selected. The aims of this study were to determine the antimicrobial resistance profile of *E. coli* isolated from stool samples of omnivores, vegetarians and vegans, to determine the phylogenetic relationship among these isolates and to detect their virulence-associated genes (VAG).

## 2. Materials and Methods

For this study, stool samples from omnivorous, vegetarian and vegan people were used. Volunteers were recruited through the dissemination of the project through social media, pamphlets and through people close to the researchers involved in the project.

To select the participants, a questionnaire was applied. Participants who followed their diet for at least six months were included in the study. Those who were hospitalized or used antibiotics in the last six months, as well as vegetarians and vegans who ate meat within the same period, were not included in the study. The control group included 60 omnivore isolates and the study group included 58 samples, 41 from vegetarians and 17 from vegans. Each member of the groups signed an informed consent form (Supplementary S1) and answered a sociodemographic questionnaire (Supplementary S2). The study was approved by the Ethics and Research Committee of the State University of Londrina CAAE 56869816.0.0000.5231.

### 2.1. Bacterial Isolates

Approximately 1 g of stool sample from each patient was inoculated into three separate tubes: one containing only 5 mL of *E. coli* (EC) broth (Merck, Darmstadt, Germany), another with 5 mL of EC broth and norfloxacin 4 μg/mL, and a third tube with 5 mL of EC broth and ceftriaxone 4 μg/mL, in order to select isolates with greater potential for antimicrobial resistance. The tubes were incubated at 44 °C in a water bath for 24 h; the temperature chosen was due to the ability of the fecal coliform group, present in warm-blooded animals such as humans, to differentiate from total coliforms by its ability to grow at $44.5 \pm 0.5$ °C [15]. After incubation, with the aid of 10 μL disposable loops, an aliquot from each tube was streaked, separately, in chromogenic agar (Becton Dickinson GmbH, Franklin Lakes, NJ, USA) and MacConkey agar (Merck, Darmstadt, Germany) and kept at 37 °C for 24 h. With the growth of colonies suggestive of *E. coli* (that grew on MacConkey and chromogenic agar, of medium to large size and with a pink to red color), about five colonies were chosen to be identified by the VITEK®2 system (bioMérieux, Paris, France), using VITEK®2 ID card. Bacterial isolates were stored both in nutrient agar (room temperature) and in Tryptic soy broth supplemented with 15% glycerol (−20 °C).

### 2.2. Antimicrobial Susceptibility

Antibiograms were performed according to the CLSI 2019 (Clinical and Laboratory Standards institute, Wayne, PA, USA, 2019), using the diffusion disc method on Mueller–Hinton agar, and interpreted also following CLSI 2019 criteria. The bacterial susceptibility was tested for 14 antibiotics: ampicillin, amoxicillin/clavulanate, ceftriaxone, cefepime, ertapenem, meropenem, nalidixic acid, ciprofloxacin, norfloxacin, gentamicin, amikacin, nitrofurantoin, trimethoprim-sulfamethoxazole and piperacillin-tazobactam.

Moreover, the disc approximation method was carried out to detect ESBL, also on Mueller–Hinton agar, using amoxicillin/clavulanic acid, ceftazidime, aztreonam, ceftriaxone and cefepime [16].

### 2.3. Detection of β-Lactamases

The detection of $bla_{CTX-M1}$, $bla_{CTX-M2}$, $bla_{CTX-M8/25}$ and $bla_{CTX-M9}$ genes was performed by conventional PCR, as described by Dallene et al. (2010) [17], and $bla_{CTX-M15}$ according to Leflon-Guibolt et al. (2004) [18]. PCR was enhanced by using the TopTaq® Master Mix Kit (QIAGEN).

Genes encoding CTX-M ESBL enzymes and their respective primers and amplicon sizes (bp) are shown in Table 1.

**Table 1.** Genes encoding CTX-M ESBL enzymes.

| Enzymes | Genes | Primer Sequences (5′–3′) | Amplion Size (bp) | References |
|---|---|---|---|---|
| CTX-M-1 | $bla_{CTX-M1}$ | TTAGGAARTGTGCCGCTGYA CGATATCGTTGGTGGTRCCAT | 688 | Dallene et al., 2010 [17] |
| CTX-M-2 | $bla_{CTX-M2}$ | CGTTAACGGCACGATGAC CGATATCGTTGGTGGTRCCAT | 404 | Dallene et al., 2010 [17] |
| CTX-M-8/25 | $bla_{CTX-M8}$ | AACRCRCAGACGCTCTAC TCGAGCCGGAASGTGTYAT | 326 | Dallene et al., 2010 [17] |
| CTX-M-9 | $bla_{CTX-M9}$ | TCAAGCCTGCCGATCTGGT TGATTCTCGCCGCTGAAG | 561 | Dallene et al., 2010 [17] |
| CTX-M-15 | $bla_{CTX-M15}$ | ATA AAA CCG GCA GCG GTGGAA TTT TGA CGA TCG GGG | 483 | Leflon-Guibout et al., 2004 [18] |

### 2.4. Detection of Virulence-Associated Genes

Genes commonly associated with virulence factors in ExPEC were investigated by PCR assay [19]. Isolates were investigated for the presence of genes encoding haemolysins (*hly*A and *hly*F), cytotoxic necrotizing factors (*cnf1* and *cnf2*), colicin V (*cva*C), aerobactin (*iut*A), yersiniabactin (*fyu*A), salmochelin (*iro*N), type 1 fimbrialadhesin (*fim*H), P-fimbriae

(*pap*C and *pap*G), S-fimbrialadhesins (*sfa*A and *sfa*S), afimbrialadhesin (*afa*), serum resistance (*iss* and *tra*T), brain microvascular endothelium invasion (*ibe*A), capsules (*kpsMT* K1, *kpsMT* K5, *kpsMT* II, and *kpsMT* III) and an outer membrane protein (*omp*T). Genes encoding these virulence factors and their respective primers and amplicon sizes (bp) are shown in Table 2 [19,20].

**Table 2.** Virulence-associated genes.

| Genes | Primer Sequences (5′–3′) | VAGs | Amplion Size (bp) | References |
|---|---|---|---|---|
| *kpsMT*II | GCG CAT TTG CTG ATA CTG TTG<br>CAT CCA GAC GAT AAG CAT GAC CA | Group 2 of capsular antigens | 272 | Johnson and Stell, 2000 [20] |
| *KpsMT*III | TCC TCTT GCT ACT ATT CCC CCT<br>AGG CGT ATC CAT CCC TCC TAA C | Group 3 of capsular antigens | 392 | Johnson and Stell, 2000 [20] |
| *KpsMT*k1 | TAG CAA ACG TTC TAT TGG TGC<br>CAT CCA GAC GAT AAG CAT GAC CA | K1 capsule | 153 | Johnson and Stell, 2000 [20] |
| *kpsMT*k5 | CAG TAT CAG CAA TCG TTC TGT A<br>CAT CCA GAC GAT AAG CAT GAC CA | K5 capsule | 159 | Johnson and Stell, 2000 [20] |
| *cva*C | CAC ACA CAA ACG GGA GCT GTT<br>CTT CCC GCA GCA TAG TTC CAT | Colicin V | 680 | Johnson and Stell, 2000 [20] |
| *iut*A | GGC TGG ACA TCA TGG GAA CTG G<br>CGT CGG GAA CGG GTA GAA TCG | Aerobactin siderophore receptor | 300 | Johnson and Stell, 2000 [20] |
| *fim*H | TGC AGA ACG GAT AAG CCG TGG<br>GCA GTC ACC TGC CC TCC GGT A | Fimbriae type 1 | 508 | Johnson and Stell, 2000 [20] |
| *fyu*A | TGA TTA ACC CCG CGA CGG AA<br>CGC AGT AGG CAC GAT CTT GTA | Yersiniobactinsiderophore receptor | 880 | Johnson and Stell, 2000 [20] |
| *pap*C | GAC GGC TGT ACT GCA GGG TGT GGC G<br>ATA TCC TTT CTG CAG GCA GGG TGT GGC | P Fimbriae | 328 | Johnson and Stell, 2000 [20] |
| *pap*G | CTG TAA TTA CGG AAG TGA TTT CTG<br>CTG TAA TTA CGG AAG TGA TTT CTG | P Fimbriae | 1070 | Johnson and Stell, 2000 [20] |
| *sfa*A | CTC CGG AGA ACT GGG TGC ATC TTA C<br>CGG AGG AGT AAT TAC AAA CCT GGC A | Sfa fimbriae | 410 | Johnson and Stell, 2000 [20] |
| *sfa*S | GTG GAT ACG ACG ATT ACT GTG<br>CCG CCA GCA TTC CCT GTA TTC | Sfa fimbriae | 240 | Johnson and Stell, 2000 [20] |
| *afa* | GGC AGA GGG CCG GCA ACA GGC<br>CCC GTA ACG CGA CAG CAT CTC | Afa fimbriae | 750 | Johnson and Stell, 2000 [20] |
| *ibe*A | AGG CAG GTG TGC GCC GCG TAC<br>TGG TGC TCC GGC AAA CCA TGC | Invasion of brain endothelium | 170 | Johnson and Stell, 2000 [20] |
| *Hly* | AAC AAG GAT AAG CAC TGT TCT GGC<br>ACC ATA TAA GCG GTC ATT CCC GTC | Hemolysin | 1177 | Johnson and Stell, 2000 [20] |
| *cnf*1 | AGG AAG TTA TAT TTC CGT AGG<br>GTA TTT GCC TGA ACC GTA A | Cytotoxic necrotizing factor 1 | 498 | Johnson and Stell, 2000 [20] |
| *cnf*2 | AAT CTA ATT AAA GAG AAC<br>CAT GCT TTG TAT ATC TA | Cytotoxic necrotizing factor 2 | 543 | Johnson and Stell, 2000 [20] |
| *tra*T | GGT GTG GTG CGA TGA GCA CAG<br>GGT GTG GTG CGA TGA GAC CAG | Serum resistance | 290 | Johnson and Stell, 2000 [20] |
| *iro*N | AAT CCG GCA AAG AGA CGA ACC GCC T<br>GTT CGG GCA ACC CCT GCT TTG ACT TT | Salmochelinsiderophore receptor | 553 | Johnson and Stell, 2000 [20] |
| *omp*T | TCA TCC CGG AAG CCT CCC TCA CTA CTA T<br>TAG CGT TTG CTG CAC TGG CTT CTG ATA C | Episomal outer membrane protease | 496 | Johnson and Stell, 2000 [20] |
| *hly*F | GGC CAC AGT CGT TTA GGG TGC TTA CC<br>GGC GGT TTA GGC ATT CCG ATA CTC AG | Putative avian hemolysin | 450 | Johnson and Stell, 2000 [20] |
| *Iss* | CAG CAA CCC GAA CCA CTT GAT G<br>AGC ATT GCC AGA GCG GCA GAA | Episomal increased serum survival | 323 | Johnson and Stell, 2000 [20] |

*2.5. Detection of Pathogenicity Island (PAI) Markers*

Different PAIs (PAI I536, II536, III536, IV536, ICFT073, IICFT073, IJ96 and IIJ96), previously characterized in uropathogenic *E. coli*, were investigated according to Sabaté et al. (2006) [21] and Koga et al. (2015) [19] and are described in Table 3.

**Table 3.** Genes encoding PAI markers.

| PAI | Primers | Sequences (5′–3′) | Amplicon Size (bp) | References |
|---|---|---|---|---|
| PAI I$_{536}$ | I.9<br>I.10 | TAA TGC CGG AGA TTC ATT GTC<br>AGG ATT TGT CTC AGG GCT TT | 1800 | Koga et al., 2014 [19] |
| PAI II$_{536}$ | orf1up<br>orf1down | CAT GTC CAA AGC TCG AGC C<br>CTA CGT CAG GCT GGC TTT G | 1000 | Sabaté et al., 2006 [21] |
| PAI III$_{536}$ | sfaAI.1<br>sfaAI.2 | CGG GCA TGC ATC AAT TAT CTT TG<br>TGT GTA GAT GCA GTC ACT CCG | 161 | Sabaté et al., 2006 [21] |
| PAI IV$_{536}$ | IRP2 FP<br>IRP2 RP | AAG GAT TCG CTG TTA CCG GAC<br>TCG GGC AGC GTT TCT TCT | 300 | Sabaté et al., 2006 [21] |
| PAI I$_{CFT073}$ | RPAi<br>RPAf | GGA CAT CCT GTT ACA GCG CGC A<br>TCG CCA ATC ACA GC GAA C | 930 | Sabaté et al., 2006 [21] |
| PAI II$_{CFT073}$ | cft073.2Ent1<br>cft073.2Ent2 | ATG GAT GTT GTA TCG CGC<br>ACG AGC ATG TGG ATC TGC | 400 | Sabaté et al., 2006 [21] |
| PAI I$_{J96}$ | papGIf<br>papGIr | TCG TGC TCA GGT CCG GAA TTT<br>TGG CAT CCC ACA TTA TCG | 400 | Sabaté et al., 2006 [21] |
| PAI II$_{J96}$ | hlyd<br>cnf | GGA TCC ATG AAA ACA TGG TTA ATG GG<br>GAT ATT TTT GTT GCC ATT GGT TAC C | 2300 | Sabaté et al., 2006 [21] |

*2.6. Phylogenetic Classification*

Seven phylogenetic groups (A, B1, B2, C, D, E and F) were considered to classify *E. coli* isolates, based on the presence of the genes *chu*A, *yja*A, *arp*A and *trp*A, as well as a DNA fragment (TSPE4.C2), all detected by PCR according to Clermont et al. (2013) [22].

Genes researched for phylogenetic classification and their respective primer sequences are described in Table 4.

**Table 4.** Genes researched for phylogenetic classification.

| PCR | Genes | Primer Sequences (5′–3′) | Amplicon Size (bp) | References |
|---|---|---|---|---|
| Multiplex | *chu*A | GAC GAA CCA ACG GTC AGG AT<br>TGC CGC CAG TAC CAA AGA CA | 279 | |
| | *yja*A | TGA AGT GTC AGG AGA CGC TG<br>ATG GAG AAT GCG TTC CTC AAC | 211 | Clermont et al., 2013 [22] |
| | TSPE4.C2 | GAG TAA TGT CGG GGC ATT CA<br>CGC GCC AAC AAA GTA TTA CG | 152 | |
| E group | *arp*A | GAT TCC ATC TTG TCA AAA TAT GCC<br>GAA AAG AAA AAG AAT TCC CAA GAG | 219 | Clermont et al., 2013 [22] |
| C group | *trp*A | AGT TTT ATG CCC AGT GCG AG<br>TCT GCG CCG GTC ACG CCC | 489 | Clermont et al., 2013 [22] |

*2.7. Statistical Analysis*

Categorical data were classified by frequencies and percentages and performed by Fisher's exact test or chi-square test, as appropriate, at an alpha significance level of 0.05.

Data analysis was performed using Statistical Package for Social Sciences (SPSS–IBM Corp., New York, NY, USA), version 20.0 for windows.

**3. Results**

*3.1. Antibiotic Resistance*

Antimicrobial resistance in *E. coli* has been reported all over the world. Studies show that the occurrence and susceptibility profile have significant differences that vary according to population, environment and geographic location [23]. In our study, there was no statistically significant difference concerning the antimicrobial resistance profile between groups. However, vegetarians/vegans had a great number of isolates resistant to beta-lactams when compared to omnivores. Isolates from both groups exhibited a high

rate of resistance to ampicillin (omnivores 80% and vegetarians/vegans 69%), amoxicillin-clavulanic acid (omnivores 32% and vegetarians/vegans 22%) and nalidixic acid (omnivores 28% and vegetarians/vegans 26%), and some of them were ESBL-positive (7–12% of isolates from vegetarians/vegans and 3–5% of isolates from omnivores).

The antimicrobial resistance profile is shown in Figure 1.

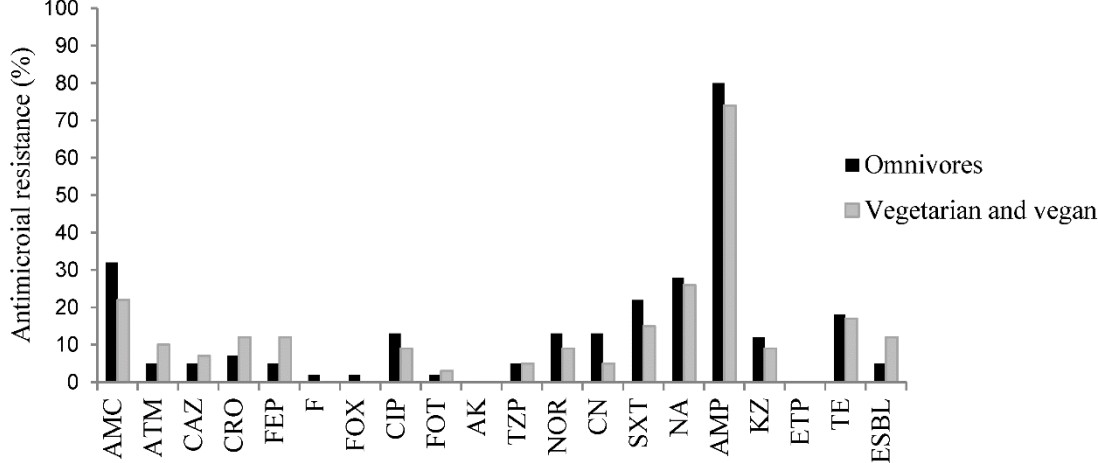

**Figure 1.** Percentage of antimicrobial resistance profile of *E. coli* isolated from omnivores and vegetarians/vegans. Amoxicillin-clavulanic acid (AMC), Aztreonam (ATM), Ceftazidime (CAZ), Ceftriaxone (CRO), Cefepime (FEP), Nitrofurantoin (F), Cefoxitin (FOX), Ciprofloxacin (CIP), Cefotaxime (FOT), Amikacin (AK), Piperacilin-tazobactam (TZP), Norfloxacin (NOR), Gentamicin (CN), Trimethoprim-sulfamethoxazole (SXT), Nalidixic acid (NA), Ampicilin (AMP), Cefazolin (KZ), Ertapenem (ETP), Tetracycline (TE) and Extended-spectrum β-Lactamase (ESBL).

Among the resistance mechanisms of ESBL-producing bacteria, the CTX-M enzyme is one of the most common and important β-lactamases described. Herein, we found that 70% of the analyzed isolates were CTX-M-9-positive, being the most prevalent in both groups (67% omnivores and 71% vegetarians/vegans). All results of CTX-M ESBLs are shown in Table 5.

**Table 5.** Prevalence of CTX-M ESBLs among *E. coli* isolated from omnivores and vegetarians/vegans.

| Genes | Omnivores *n* (%) | Vegetarians/Vegans *n* (%) |
|---|---|---|
| CTX-M-1 | - | 4 (57) |
| CTX-M-2 | 1 (33) | 3 (43) |
| CTX-M-8/25 | - | - |
| CTX-M-9 | 2 (67) | 5 (71) |
| CTX-M-15 | - | 2 (29) |
| Total of isolates | 3 | 10 |

### 3.2. Virulence-Associated Genes

The most predominant VAGs among isolates from omnivores were *fim*H (70%), *iut*A (32%), *fyu*A (32%) and *tra*T (32%), while among isolates from vegetarians/vegans, the most predominant were *tra*T (62%), *kpsMT* k1 (28%) and *iut*A (22%).

Moreover, there was a statistically significant difference between control and study groups in relation to isolates that were *kpsMT* III-positive. This gene was detected in nine isolates from vegetarians/vegans and it was not detected in isolates from omnivores. In addition, 42 isolates from omnivores were *fim*H-positive, while only seven isolates from vegetarians/vegans presented this gene. Regarding *cnf* 1, which codes for a necrotizing cytotoxic factor, it was detected in four isolates from vegetarians/vegans and it was not

detected in isolates from omnivores. Finally, 36 isolates from vegetarians/vegans were *tra*T-positive, while this gene was detected in 19 isolates from omnivores.

The results for the VAGS are described in Table 6.

**Table 6.** Prevalence of VAGs among *E. coli* isolated from omnivores and vegetarians/vegans.

| Virulence-Associated Genes | Coding For | Omnivores $n = 60$ (%) | Vegetarians Vegans $n = 58$ (%) | *p*-Value |
|---|---|---|---|---|
| *kpsMT* II | Group 2 of capsular antigens | 15 (25) | 10 (17) | 0.349 |
| *kpsMT* III | Group 3 of capsular antigens | 0 | 9 (15) | 0.001 * |
| *kpsMT* k1 | K1 capsule | 9 (15) | 16 (28) | 0.078 |
| *kpsMT* k5 | K5 capsule | 14 (23) | 8 (14) | 0.214 |
| *cva*C | Colicin V | 3 (5) | 7 (12) | 0.152 |
| *iut*A | Aerobactinsiderophore receptor | 19 (32) | 13 (22) | 0.308 |
| *fim*H | Fimbriae type 1 | 42 (70) | 7 (12) | <0.001 * |
| *fyu*A | Yersiniobactinsiderophore receptor | 19 (32) | 11 (19) | 0.206 |
| *pap*C | P Fimbriae | 7 (12) | 7 (12) | 0.894 |
| *pap*G | P Fimbriae | 2 (3) | 2 (3) | 0.946 |
| *sfa*A | Sfa fimbriae | 6 (10) | 4 (7) | 0.582 |
| *sfa*S | Sfa fimbriae | 4 (7) | 2 (3) | 0.451 |
| *afa* | Afa fimbriae | 2 (3) | 2 (3) | 0.946 |
| *ibe*A | Invasion of brain endothelium | 5 (8) | 3 (5) | 0.526 |
| *hly* | Hemolysin | 4 (7) | 8 (14) | 0.180 |
| *cnf*1 | Cytotoxic necrotizing factor 1 | 0 | 4 (7) | 0.035 * |
| *cnf*2 | Cytotoxic necrotizing factor 2 | 3 (5) | 4 (7) | 0.631 |
| *tra*T | Serum resistance | 19 (32) | 36 (62) | 0.001 * |
| *iro*N | Salmochelinsiderophore receptor | 8 (13) | 9 (15) | 0.682 |
| *omp*T | Episomal outer membrane protease | 4 (7) | 9 (15) | 0.110 |
| *hly*F | Putative avian hemolysin | 2 (3) | 6 (10) | 0.118 |
| *Iss* | Episomal increased serum survival | 12 (20 | 3 (5) | 0.019 * |

* indicates significant difference.

### 3.3. PAIs

Most isolates from omnivores (55%) were PAI I$_{536}$-positive($\alpha$-hemolysin, CS12 fimbriae and F17-like fimbriae adhesin) [21,24], while most isolates from vegetarians/vegans (59%) were PAI IV$_{536}$-positive (yersiniabactin siderophore system). Moreover, seven (12%) isolates from omnivores were positive for PAI III$_{536}$ (S-fimbriae, salmochelin, HmuR-like heme receptor, Sat toxin, Tsh-like hemoglobin protease, antigen 43), which was not detected in isolates from vegetarians/vegans.

Data related to the prevalence of PAIs of isolates are shown in Table 7.

**Table 7.** Prevalence of PAIs among *E. coli* isolated from omnivores and vegetarians/vegans.

| Genes | Omnivores $n = 60$ (%) | Vegetarians/Vegans $n = 58$ (%) | *p*-Value |
|---|---|---|---|
| **PAI** | | | |
| PAI I$_{536}$ | 33 (55) | 7 (12) | <0.001 * |
| PAI II$_{536}$ | 0 | 0 | |
| PAI III$_{536}$ | 7 (12) | 0 | 0.008 * |
| PAI IV$_{536}$ | 23 (38) | 34 (59) | 0.018 * |
| PAI I$_{CFT073}$ | 6 (10) | 8 (14) | 0.483 |
| PAI II$_{CFT073}$ | 11 (18) | 9 (15) | 0.744 |
| PAI I$_{J96}$ | 0 | 3 (5) | 0.070 |
| PAI II$_{J96}$ | 3 (5) | 0 | 0.090 |

* indicates significant difference.

### 3.4. Phylogeny

Phylogenetic group A, which contains most of the commensal non-pathogenic isolates that survive in the gastrointestinal system, was the most prevalent in both groups, and no statistically significant difference was found between them.

Data related to the phylogenetic classification of isolates are shown in Table 8.

**Table 8.** Phylogenetic classification among *E. coli* isolated from omnivores and vegetarians/vegans.

| Genes | Omnivores *n* = 60 (%) | Vegetarians/Vegans *n* = 58 (%) | *p*-Value |
|---|---|---|---|
| **Phylogenetic Classification** | | | |
| A | 15 (25) | 13 (22) | 0.818 |
| B1 | 8 (13) | 10 (17) | 0.506 |
| B2 | 4 (7) | 11 (19) | 0.079 |
| C | 2 (3) | 2 (3) | 0.703 |
| D | 12 (20) | 8 (14) | 0.414 |
| E | 9 (15) | 10 (17) | 0.506 |
| F | 10 (17) | 4 (7) | 0.073 |

## 4. Discussion and Conclusions

This study evaluated the virulence and antibiotic resistance profile in *E. coli* from feces of individuals following vegan, vegetarian and omnivorous diets. Isolates from both groups, vegetarians/vegans and omnivores, harbored genes related to ESBLs and exhibited a high rate of resistance to first-choice antimicrobials used in the treatment of UTIs.

In both groups, most of ESBL-positive isolates were positive for CTX-M-9 (two isolates among omnivores and five among vegetarians/vegans). Presently, CTX-M ESBLs include more than 220 different enzymes clustered into five subfamilies (CTX-M-1, CTX-M-2, CTX-M-8, CTX-M-9 and CTX-M-25) based on their amino acid identities, and enzymes that originated from subfamilies, such as CTXM-1 and CTX-M-9, which are widely distributed and commonly reported [25]. In general, CTX-M enzymes are more active against cefotaxime and ceftriaxone than ceftazidime, but point mutations involving the active site of some enzymes, especially those belonging to the CTX-M-1 and CTX-M-9 subfamilies, have significantly increased their ability to hydrolyze ceftazidime [26].

As we undergo antibiotic treatments many times throughout our lives, bacterial microbiota are exposed to selective pressure by antibiotics. Consequently, the gastrointestinal tract is highly exposed, especially during oral therapy, and this results in a natural selection of resistant isolates carrying an important genetic pool that might be capable of transferring their antibiotic resistance genes to other isolates present in the human intestine. Moreover, resistant food contaminants, such as organic fertilizers, originated from animals and consumed by humans, can also act as a gene pool (donors) of antibiotic resistance genes [8].

In agriculture, organic fertilizers, such as manure, are conventionally used to improve crop yield, since they are believed to be more effective than inorganic fertilizers. However, because of the indiscriminate use of antibiotics and the high spread of antibiotic resistance genes (ARGs), manure is regarded as an important food contaminant, since it may contain microorganisms harboring these ARGs, which pose a high risk to human health [27]. Animal manure is an important reservoir of antibiotic residues and ARGs, in addition to human pathogenic bacteria [28]. Perhaps for this reason, a greater number of ESBL-positive isolates was found among vegetarians and vegans (10) when compared to omnivores (3).

Furthermore, after the report of the first cases of antibiotic-resistant bacterial diseases in humans, recommendations were made for banning the use of antibiotics as growth promoters if the same drugs are also prescribed in human medicine [29]. However, recently, it was estimated that approximately 3,345,022 kg of antimicrobials were sold and used in the U.S. poultry industry in 2016, with 1,265,420 kg being "medically important" for human medical therapy [29,30]. Due to this, there is growing interest in sustainable food production, and research is currently being conducted to identify antibiotic alternatives that could support healthy growth and provide defense against pathogenic microbes [30].

Most *E. coli* isolates in both groups belong to phylogenetic group A, shared by *E. coli* isolates commensally inhabiting the mucosa of the gastrointestinal tract [31]. The next phylogenetic groups most found among isolates from vegetarians/vegans and omnivores were B2 and D, respectively. According to Sarowska et al. (2019) [32], B2 and D are often related to extra-intestinal infections. These results are in accordance with previous studies, which show that, despite being pathogenic, phylogenetic group B2 has a high prevalence ($\geq 25\%$) in fecal *E. coli* [6,33] and is also related to ExPEC in the human intestinal microbiota.

Interestingly, the groups showed significant differences in relation to the presence of VAGs. Among omnivores, 70% (42) of the isolates harbored *fim*H, which codes for an adhesin responsible for supporting the binding of bacteria to specific host cells in structural or functional molecules associated with the cell membrane [34]. On the other hand, 62% (36) of isolates from vegetarians/vegans harbored *tra*T, carried by conjugative plasmids, which favors its transmission. In addition, *tra*T is associated with an increased serum resistance, observed mainly in ExPEC present in the bloodstream, and among commensal isolates, *tra*T is a good predictor of urinary tract and bloodstream infections. The fimbriae play a significant role in the process of adhesion of the microorganism to the cells, increasing the virulence of pathogenic *E. coli* due to this close contact of the bacteria with the host cell wall. Most genes that determine the presence of fimbriae on the surface of bacterial cells are chromosomally encoded or, less frequently, within plasmid DNA, which facilitates their horizontal transfer between different isolates [32,35].

The pathogenicity island most frequent (55%) among isolates from omnivores was PAI I$_{536}$, while only 12% (7) of isolates from vegetarians and vegans presented it. Remarkably, PAI IV$_{536}$ was detected in 59% (36) and 38% (23) of isolates from vegetarians/vegans and omnivores, respectively. This PAI, which codes for the *fyuA* gene (yersiniabactin iron absorption system), is present in up to 57% of *E. coli* commensal isolates and is also the most frequent in enterobacteria.

Some VAGs were found in only one of the groups, such as PAI III$_{536}$, found in 12% (7) of the isolates from omnivores, while the *kpsMT* III gene was detected only among 15% (9) of isolates from vegetarians/vegans. Interestingly, this gene codes for a polysaccharide capsule found mainly in *E. coli* isolates causing intestinal infections, including EPEC, ETEC and EHEC [36].

Regarding *cnf*1, it was detected in 7% (4) of isolates from omnivores. This gene codes for a necrotizing cytotoxic factor capable of facilitating bacterial internalization in host cells and contributing to the potential for invasion of pathogenic *E. coli* by manipulating epithelial and endothelial barriers [37]. It is often associated with extraintestinal conditions, especially urinary tract infections, present in around 40% of UPEC isolates, and less frequently, in bacteremia and neonatal meningitis [32].

A study by Chen et al. (2020) [38] shows that vegetarian diets (compared to a non-vegetarian diet) are associated with a lower risk of UTIs, especially in women, non-smokers and for the uncomplicated UTI subtype. In addition, other previous studies have suggested that meat-related food-based ExPECs may be the main pathogens in uncomplicated UTIs. However, despite the differences found between omnivores and vegans/vegetarians, it is not possible to conclude that one diet is better than the other, since the composition of these diets in terms of nutrients does not seem to be sufficient for them to shape the intestinal microbiota.

The present study also shows that even commensal *E. coli* isolated from the feces of healthy individuals have great potential to cause extraintestinal infections. This is due not only to the high prevalence of VAGs among these isolates, but also to the presence of mobile genetic elements capable of directly influencing the genomes of pathogenic bacteria and the horizontal transfer of resistance genes, which can spread rapidly among non-pathogenic isolates, transforming them into potential pathogens that can cause UTIs and several other extraintestinal infections.

**Supplementary Materials:** The following supporting information can be downloaded at: https://www.mdpi.com/article/10.3390/microbiolres13040058/s1.

**Author Contributions:** E.C.V., Z.N.T. and R.K.T.K. conceived and designed the experiments. A.T.T. and J.G.M.S. performed the experiments. E.C.V., J.G.M.S., V.L.K. and E.C.V. analyzed the data. E.C.V., Z.N.T. and R.K.T.K. contributed reagents and materials. E.C.V. and Z.N.T. reviewed the study. V.L.K. performed confirmation experiments. All authors have read and agreed to the published version of the manuscript.

**Funding:** The support was provided by the Bill and Melinda Gates Foundation's Grand Challenges Explorations Brazil—New Approaches to Characterize the Global Burden of Antimicrobial Resistance (number OPP1193112). No funding agencies had any role in this study's design, collection, analysis, data interpretation and manuscript writing.

**Data Availability Statement:** Not applicable.

**Acknowledgments:** We are grateful to the graduate program in clinical and laboratory pathophysiology, to the National Council for Scientific and Technological Development for the master's scholarship awarded to Ariane Tiemy Tizura, and to Danilo Figueiredo for his help in publicizing the project.

**Conflicts of Interest:** The authors declare that the research was conducted in the absence of any commercial or financial relationships that could be construed as a potential conflict of interest.

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
