# Peer review of "Escherichia coli Isolated from Vegans, Vegetarians and Omnivores: Antibiotic Resistance, Virulence Factors, Pathogenicity Islands and Phylogenetic Classification"

_2036-7481, doi:10.3390/microbiolres13040058_

Round 1

Reviewer 1 Report (Previous Reviewer 3)

The authors have significantly improved the text of the revised manuscript, but it is necessary to work more carefully on the references: number them in the text and put in square brackets.

Author Response

Dear reviewer, thank you for the report. We made the requested changes in the paper. Please consider them.

Reviewer 2 Report (Previous Reviewer 2)

1. Introduction still needs improving. Again, why do the authors focus their introduction on UPEC and ExPEC? That needs to be explained. Also, the authors talk about virulence factors found in EPEC, ETEC and EHEC (line 29), but no information whatsoever is given about those pathotypes.

2. In this study the authors take samples of up to 118 individuals, but without giving information of the location of where those individuals live. In line 82, authors talk about samples from “people of Brazil”, but as they know better than me, Brazil is a big country and really diverse. This information should be provided.  

3. As previously discussed, other factors apart from the diet will affect the intestinal microbiota, independently if they eat meat or not. As previously said, 72% of the vegetarian individuals sampled here, own pets and are in close contact with animals, modifying its microbiota. Also, neither the vegetarian nor the omnivorous diet seems to be defined. What’s the fat and sugar content? Are they having an equilibrate diet? Do the subjects cook their own food?  With the given information, I do not believe that the authors can really separate between the “study group” (vegetarians) and the “control group”. A more controlled and rigorous study is required for that. Moreover, I would recommend the authors to give a look at the following study: DOI: 10.1080/10408398.2019.1676697; where the authors do not found a consistent association between a vegan diet or vegetarian diet and microbiota composition compared to omnivores.

Author Response

1 e 2. Dear reviewer, we made the improvements requested in the introduction. Please consider our changes to the article. Thank you.

3. Dear reviewer, thank you very much for your considerations and indication of the paper "Is a vegan or a vegetarian diet associated with the microbiota composition in the gut? Results of a new cross-sectional study and systematic review" which really brings important considerations until they conclude that not found a consistent association between a vegan diet or vegetarian diet and microbiota composition compared to omnivores. When we designed the study, it was only thought to see if
E. coli from omnivores was different from vegans/vegetarians, regarding its resistance and virulence. But your considerations about the patient's diet are very relevant. It will certainly be included in future studies.

Reviewer 3 Report (New Reviewer)

This is a very interesting manuscript that I read with pleasure. The authors performed characterization of E. coli isolated from two groups of people (omnivores : vegans and vegetarians) and present in their manuscript some quite interesting results. I have just some minor comments:

Title: please include the word “antibiotic” in front of “resistance”.

Throughout the manuscript, when you state/associate with your PCR-virulence-gene data use “virulence-associated genes” or “VAG” instead of “virulence factors”.

Material and methods:

-line 92 – change “semester” for “6 months” as was done in line 90

-section Bacterial Isolates – please provide a more extended description of isolating the bacterial isolates and provide information whether screening for non-clonality was performed (in the case not, take care that throughout the manuscript the word “isolate” is used instead of “strains”. Further, check the temperature that was used in the water bath – was it really 44 °C. If yes, provide an explanation, as this is unusual.

-line 132 – “investigated for the presence of” should be changed into “investigated for the presence of genes encoding”

Results:

To start the Results section with: “There was no statistically significant…” is a to sharp beginning of this section. Please add some introductory sentence. 

Divide the Results section into subtitles for different types of results (Antibiotic resistance - Virulence-associated genes – PAIs – Phylogeny).

Dived the large Table 6 (VAG, PAIs and Phylo data) into separate tables. In the table presenting VAG, change the order of columns, so that in the first column it will be “Virulence-associated gene” and in the second “Coding for”

Discussion and Conclussion:

Lines 236-237: the sentence starting with “Moreover…” is a bit confusingly written. Please reformulate it.

In this section, please include the paper PMID: 25657191, DOI: 10.1093/femsle/fnu061, as it nicely fits into the manuscript.

Author Response

Dear reviewer, thank you for your comments. We made the necessary changes throughout the paper. I ask you to consider them, please. 

Round 2

Reviewer 2 Report (Previous Reviewer 2)

The authors improved the introduction. Still, they do not provide a proper location. Instead they added “a city in Southern Brazil” (Line 86). Which one? In materials and methods they say that it is “close to the researchers involved in the project”. Is this mysterious city Londrina? Then say it.  

As said before, the study is biased and the definition of groups (vegetarians vs omnivorous) was not properly done. Other factors were not taken into consideration and the study has serious limitations; but none of those have been addressed after several rounds of revision or even mentioned in the discussion section. I am sorry, but from my point of view, this manuscript should not be published as it is. I would recommend the authors to change it completely and transform it into a description of the E. coli found in the members of the University of Londrina, instead of trying to compare between groups. But for that, the manuscript needs more than just cosmetic changes.

Author Response

Point 1: The authors improved the introduction. Still, they do not provide a proper location. Instead they added “a city in Southern Brazil” (Line 86). Which one? In materials and methods they say that it is “close to the researchers involved in the project”. Is this mysterious city Londrina? Then say it. 

Response 1: Dear reviewer, when we refer in the paper "volunteers were recruited through the dissemination of the project through social media, pamphlets and through people close to the researchers involved in the project". We meant that people close to the project's researchers also helped to recruit volunteers to participate. As we had exclusion criteria for obtaining the samples, it was not easy to recruit interested parties.

We did not include the city because the materials and methods say "The study was approved by the Ethics and Research Committee of the State University of Londrina CAAE 56869816.0.0000.5231." However, if you feel you need to put it, we can add: “a city in Southern Brazil, Londrina, in the north of Parana, Brazil”.

Point 2: As said before, the study is biased and the definition of groups (vegetarians vs omnivorous) was not properly done. Other factors were not taken into consideration and the study has serious limitations; but none of those have been addressed after several rounds of revision or even mentioned in the discussion section. I am sorry, but from my point of view, this manuscript should not be published as it is. I would recommend the authors to change it completely and transform it into a description of the E. coli found in the members of the University of Londrina, instead of trying to compare between groups. But for that, the manuscript needs more than just cosmetic changes.

Response 2: Dear reviewer, we attach the questionnaire carried out with the people who joined the project. We also forwarded the answer table, unfortunately one of the reviewers requested that it be removed from the text of the paper, but we can send it as an attachment, along with the questionnaire.

When the participants joined the project, 50% of people who are self-employed professionals (Complete higher education) were awakened to participate in the project in restaurants and another group of studying higher education from the Universidade Estadual de Londrina but also formed by students from other private universities in London. At no time was the study biased with only people from the State University of Londrina. Thank you very much for your considerations, which are very important to avoid making the same mistakes in future projects. If you need any further clarification I am available.

This manuscript is a resubmission of an earlier submission. The following is a list of the peer review reports and author responses from that submission.

Round 1

Reviewer 1 Report

The manuscript has been greatly improved now, results and discussion arguments are better présented.

Pay only attention the citations not always cited in the same manner!

Reviewer 2 Report

The introduction needs to be improved. The authors focus their introduction on UPEC and ExPEC, but then they talk about gut microbiota (line 68) and virulence factors found in EPEC, ETEC and EHEC (line 29). Also, there are several (and more up-to-date) references in the effect of diet over intestinal microbiota or transfer of antibiotic resistant genes; I would strongly recommend to the authors to give it a look.

For me the main concern persists: this study is biased. Up to a 77% of the samples belong to people that completed or are attending higher education. Is this a representative sample of the population of Brazil (line 72-74)? Any sociodemographic information that could be extracted from a biased study is not useful or worth publishing. Other variants have not been taken into consideration. What’s the age range? What’s their socioeconomical status? Do the subjects live alone? Do they cook their own food or someone does it for them? How often do they go to restaurants? Is it an equilibrated diet or is it all junk food? All those things will affect the intestinal microbiota independently if they eat meat or not. Moreover, 72% of the vegetarian group own pets and are in close contact with animals, modifying its microbiota and rendering it impossible to take conclusions out of it.

The authors did a big amount of work processing the samples, but as it is, I don’t think that they can really separate between the microbiota of vegetarians and omnivores. A recommendation would be to transform the manuscript into a description of the virulence factors found in the study cohort, but they cannot extract any conclusions out of it or differentiate between vegetarians or omnivores (that would require resampling, better collection of information and a more diverse sample group).

Reviewer 3 Report

The manuscript of A.T. Tizura et al. titled "Escherichia coli isolated from vegans, vegetarians, and omnivores: virulence factors, pathogenicity islands, and phylogenetic classification" is devoted to the comparison of virulence factor genes, pathogenicity islands, and phylogenetic classification of E. coli isolated from two groups - omnivores and vegans/vegetarians. The goal of the paper is unclear and the authors did not find any significant differences. The conclusion of the manuscript also did not show clear results on which diet is better. Since the manuscript was written by non-English-speaking authors, I would highly recommend significant professional editing of the English.

Major issues include:

The sampling size is low to make any significant conclusions. No information on how control and study groups were formed. No information on the time, place, and other backgrounds of persons who participated in the study. Are all of them were healthy? How it was checked? Which accompanying illnesses do they have?
Strains were stored at room and -20 temperature which is not enough to store. Usually, strains must be stored at -70C.
Many of the papers on the same topic are not cited.
Lines 145-155 and table 5 are irrelevant and should be omitted.

Minor issues and typos:

Format all references as numbers in square brackets as written in the Instruction for Authors.
Also, most of the references cited in the text are not listed in the References list!
Line 12: Full name for Escherichia coli
Lines 20-21: Italicize genes
Line 41: Typo: genes
Line 81: I suggest omitting "one semester"
Line 91: Which volume of media was used for isolation?
Line 91: Manufacturer of EC broth?
Line 94: Why incubation at 44 C in a water bath was used?
Line 94: Agar is solid media, so colonies were streaked, not inoculated. How many colonies from one plate/sample were used? How E. coli were identified? Why serogroups were not identified? Which O and H antigens were in isolates?
Line 95: Manufacturer of McConkey agar?
Line 100: Which antibiotics were used for the disc diffusion test?
Line 108: By saying simplex PCR did you meant conventional PCR?
Line 109: Put R in circle and superscript.
Table 1: Add "Amplicon size, bp". For some primers is not clear which genes are they.
Lines 117-122: Italicize genes
Line 123: No Koga et al. 2015 primers in Table 2?
Table 2: Omit dot in 1.070 and 1.177
Figure 1: Significance for AMR profiles?
Table 6: Add 0 or - to empty cells. Also why 1 is 33%, 2 is 67%, 4 is 57%, and so on?
Line 241: Most commensal E. coli belong to phylogroup A is a well-known fact. Add reference.
Line 261: I suggest replacing plasmidial with plasmid